# Fetal Isolated Single Umbilical Artery (ISUA) and Its Role as a Marker of Adverse Perinatal Outcomes

**DOI:** 10.3390/jcm13247749

**Published:** 2024-12-18

**Authors:** Ana María Cubo, Alicia Moreno, Mercedes Sánchez-Barba, María Ángeles Cabrero, Tatiana Costas, María O Rodríguez, María Estrella Hernández Hernández, Polán Ordás, Ana Villalba Yarza, Francisco Javier Goenaga, María Victoria Lapresa-Alcalde

**Affiliations:** 1Department of Obstetrics and Gynecology, Hospital Universitario de Salamanca, 37007 Salamanca, Spain; macabrero@saludcastillayleon.es (M.Á.C.); tcostas@saludcastillayleon.es (T.C.); morodriguezm@saludcastillayleon.es (M.O.R.); ehernandezhe@saludcastillayleon.es (M.E.H.H.); pordax@saludcastillayleon.es (P.O.); avillalba@saludcastillayleon.es (A.V.Y.); javiergoenaga@saludcastillayleon.es (F.J.G.); 2Faculty of Medicine, University of Salamanca (USAL), 37007 Salamanca, Spain; amorenorobas@usal.es (A.M.); mersanbar@usal.es (M.S.-B.); mvlapresa@saludcastillayleon.es (M.V.L.-A.); 3Institute of Biomedical Research of Salamanca (IBSAL), 37007 Salamanca, Spain; 4Department of Obstetrics and Gynecology, Hospital Virgen de la Concha, 49022 Zamora, Spain

**Keywords:** single umbilical artery, isolated single umbilical artery, aneuploidy markers, prenatal diagnosis, ultrasound markers, soft markers, umbilical cord vessels

## Abstract

Single umbilical artery (SUA) is considered an ultrasound marker of anomalies. Although it may be present in about 0.5% to 6% of normal pregnancies, it has been linked with an increased risk of fetal growth restriction (FGR), as well as cardiac, genitourinary and gastrointestinal malformations and chromosomal anomalies such as trisomies 21 and 18. **Objectives:** This study aims to evaluate whether the presence of isolated SUA (ISUA) is associated with adverse perinatal outcomes. **Methods:** A descriptive, observational and retrospective study was conducted, analyzing 1234 pregnancies (1157 normal gestations with a three-vessel cord and 77 cases of ISUA). **Results:** ISUA was associated with a lower gestational age (38 vs. 39 weeks) and a lower birth weight (3013 vs. 3183 g) when performing a univariate analysis. However, after performing a multivariate analysis adjusted for maternal age and BMI, the association between single umbilical artery (SUA) and lower birth weight could not be proven. No significant differences were found in the rate of malformations, genetic disorders, Apgar score, pH at birth or admissions in the neonatal ICU. **Conclusions:** ISUA is associated with a lower birth weight but does not increase the risk of prematurity or low-birth-weight-related neonatal admissions. Additionally, ISUA is not significantly associated with a lower gestational age, genetic disorders, fetal malformations, worse Apgar scores or lower pH values at birth.

## 1. Introduction

A prenatal examination of the umbilical cord is an essential part of the ultrasound scan in pregnancy. The presence of a single artery, known as a single umbilical artery (SUA) (Figure 1), is considered a risk marker for trisomy and may be associated with fetal malformations in up to 11–30% of cases [1,2,3]. When a single umbilical artery is observed without any other detectable malformations, it is termed isolated single umbilical artery (ISUA). The findings in existing studies are inconsistent: some studies report that the presence of SUA is linked to an increased risk of preterm birth and fetal growth restriction (FGR) and higher perinatal morbidity and mortality [2,4,5,6,7,8]. However, these associations have not been corroborated by other research [9,10], leading to conflicting conclusions regarding the implications of SUA.

The primary objective of this study was to evaluate the role of isolated single umbilical artery (ISUA) as a marker of adverse perinatal outcomes, including prematurity, low birth weight, low Apgar score, low postnatal pH, need for admission to a neonatal intensive care unit (NICU) and perinatal death.

## 2. Materials and Methods

This is a descriptive, observational and retrospective study with a total of 1157 consecutive singleton pregnant patients treated at the Department of Gynaecology and Obstetrics of the Hospital Universitario de Salamanca, during the period from 1 January to 31 August 2023. Inclusion criteria were women with singleton gestations without other associated ultrasound markers, nuchal translucency less than 3.5 mm and no fetal malformations. Exclusion criteria were multiple gestations or those that did not meet the inclusion criteria. These pregnancies were compared with 77 gestations with ISUA, whose data were collected during the years 2019 to 2023 at the same center, ISUA being defined as the presence of a single umbilical artery without any other ultrasound signs or markers or fetal malformations (Figure 2). All ultrasound examinations were performed by one of the five specialists belonging to the Prenatal Diagnosis Department at the 20th-week ultrasound scan (AMC, MOMR, PO, AVY, FJG), following the ISUOG Protocol for the performance of the routine mid-trimester fetal ultrasound scan [11]. Once ISUA was diagnosed, it was confirmed by another of these specialists in the department at a subsequent visit. This study was approved by the Hospital Ethics Committee. Considering that this was a retrospective study, no informed consent was needed from the patients to carry out this study. Sample size was calculated using the WinEpi calculator (http://www.winepi.net/f102.php, accessed on 1 June 2022), considering a prevalence of 1% for ISUA [8] with a confidence level of 95%. Data were collected in a database created specifically for this study with Microsoft Excel (Microsoft Corporation, Redmond, WA, USA), which was accessible from the outpatient clinic, the delivery room and the hospital ward.

For statistical analysis, SPSS software (IBM SPSS Statistics, version 28) was used. A normality goodness-of-fit test was performed by using the Kolmogorov–Smirnov test, finding that the quantitative variables did not follow a normal distribution, so non-parametric tests were used to analyze the results (chi-square test for qualitative variables and Mann–Whitney U test). The statistical significance level was set at 95% (*p* < 0.05).

To analyze the relationship between variables and exclude any as potential confounding factors, a multivariate logistic regression analysis adjusted for maternal age and BMI (Body Mass Index) was performed. Gestational age at birth, newborn weight, pH and Apgar scores at 1 and 5 min were included as covariates. Odds ratio, 95% confidence intervals and statistical significance (*p*) were calculated for each variable. 

## 3. Results

During the study period, a total of 1234 pregnancies (1157 with a three-vessel cord and 77 with ISUA) that met the inclusion criteria for this study were analyzed. The mean gestational age at delivery was 38 weeks. The mean maternal age was 32 years, ranging from 14 to 47 years. Most pregnancies were spontaneous (89.4%). The main mode of delivery was normal vaginal birth (58.3%), followed by caesarean section (24.7%). Induced labor occurred in 46% of gestations. Only 29 newborns out of 1234 (2.4%) needed ICU admission, all of them in the three-vessel cord group. Among all the reasons for being admitted to the neonatal unit, prematurity was the most common (46%), followed by respiratory distress, which reached 19%. Six neonatal deaths (0.5%) occurred in the immediate neonatal period, all in the control group. Data are shown in Table 1 and Table 2.

A univariate analysis of the main variables studied was performed (Table 1 and Table 2). ISUA was associated with both a lower gestational age and lower birth weight (Table 2). However, this reduction in gestational age was not linked to an increased rate of prematurity (38 vs. 39 weeks at birth). Furthermore, no association was observed between the presence of ISUA and the occurrence of genetic disorders or neonatal malformations. Similarly, ISUA was not significantly associated with poorer Apgar scores, lower pH values, increased neonatal admissions or perinatal deaths (Table 1 and Table 2).

To ensure that the lack of observed differences was not attributable to potential confounding variables, a multivariate logistic regression analysis adjusted for maternal age and BMI was performed (Table 3). Gestational age at birth and newborn weight demonstrated marginally significant associations with the presence of a single umbilical artery (*p* = 0.05). However, the confidence interval for gestational age includes 1 (95% CI: 0.99–1.63), indicating that this association may not be robust. Similarly, while the association for newborn weight reached the threshold for statistical significance (*p* = 0.05), its effect size was minimal (OR: 0.99). In contrast, pH at birth and Apgar scores at the first and fifth minutes did not show statistically significant associations with the presence of a single umbilical artery.

## 4. Discussion

Single umbilical artery (SUA) has been widely reported as a marker not only for aneuploidy and genetic disorders but also for fetal malformations [1,12], low birth weight and neonatal mortality [2,4,5,7]

In our study population, the presence of SUA was significantly associated with a lower birth weight in both the univariate and multivariate analyses. This finding aligns with most of the existing literature. For instance, the meta-analysis by Kim et al. [7] demonstrated that pregnancies with SUA have a threefold increased risk of low birth weight compared to singleton pregnancies with a three-vessel cord, with these babies weighing approximately 200 g less on average. Similarly, Siargkas et al., in a prospective study of 6528 patients, identified a significant association between SUA, lower birth weight and reduced gestational age at delivery [8]. According to their findings, fetuses with SUA were twice as likely to have low birth weight and preterm delivery as those with normal umbilical cords. Conversely, Wiegand et al. [13] argued that SUA does not increase the risk of low birth weight or preterm delivery. However, their study included 273 SUA cases without a control group, and their results were compared solely with data from the literature. To date, three meta-analyses have examined the impact of SUA on preterm birth and birth weight, yielding conflicting conclusions. While Dagklis et al. and Kim et al. reported a significant association between SUA and these outcomes, Voskamp et al. did not find evidence supporting such a relationship [4,7,10].

Our results are consistent with most of the existing literature: in our study, fetuses with SUA had a birth weight approximately 10% lower compared to those with a three-vessel umbilical cord. However, even though univariate analysis demonstrated the association of ISUA with a mean gestational age one week shorter, this could not be confirmed in multivariate analysis; despite reaching statistical significance, the confidence interval included 1, which suggests that there is insufficient statistical certainty to confirm an association between the presence of single umbilical artery (SUA) and lower gestational age. In fact, this reduction in gestational age did not increase the preterm birth rate, adhering to the WHO definition of preterm birth as delivery before 37 weeks of gestation [14].

Another significant aspect of the presence of SUA in the literature is its potential as a marker for genetic abnormalities or fetal malformations. In our study, however, the presence of SUA was not associated with genetic disorders, as none of the newborns were diagnosed with any genetic condition. Likewise, we could not establish any correlation between SUA and malformative conditions, as no such conditions were found in the newborns, which contrasts with the findings reported in other published studies [1,2,12,15]. Many studies linking SUA to genetic or structural abnormalities do not treat it as an isolated marker, as fetuses with SUA often exhibit additional malformations or associated markers [1,3,12]. In our study, we specifically focused on cases where SUA was the sole marker, excluding pregnancies with other malformations or associated markers, with the aim of investigating SUA as a single indicator. Similarly, Granese et al., in their analysis of ISUA without other markers or malformations, found no association between SUA and increased rates of aneuploidy during gestation [3].

There is limited research linking the presence of isolated SUA (ISUA) with immediate postpartum well-being parameters, such as the Apgar score and umbilical cord blood pH. The meta-analysis by Dagklis et al. and the study by Ebbing et al. report an increased risk for fetuses with SUA to exhibit lower pH values at birth and reduced Apgar scores at 5 min, as well as a heightened risk of admission to a neonatal intensive care unit and neonatal mortality [2,4]. Consequently, these authors emphasize the clinical importance of prenatal SUA assessment. Conversely, our study did not replicate these findings, as no statistically significant differences were observed in any of these parameters either in univariate or multivariate analysis. None of the neonates with ISUA required admission to the neonatal unit, and no perinatal deaths were recorded. This discrepancy is likely attributable to the characteristics of the study populations. The referenced studies included gestations with fetuses presenting multiple anomalies, where SUA was not analyzed as an isolated finding but rather as part of a broader syndrome. In contrast, our study exclusively included pregnancies with isolated SUA, where all other parameters were normal, suggesting that these populations are not directly comparable.

## 5. Strengths and Limitations

The main strength of this study lies in the analysis of single umbilical artery (SUA) as an isolated marker. Cases were carefully selected to include only those where SUA was present without concurrent conditions, allowing for an assessment of its specific influence without confounding effects from other markers or findings. This approach is particularly noteworthy, as most studies in the literature examine SUA in conjunction with multiple conditions, making it challenging to delineate the distinct impact of this marker. Additionally, the analysis extended beyond neonatal admission to include multiple neonatal variables, such as fetal pH, low birth weight and Apgar scores. This comprehensive approach provides a more detailed evaluation, enabling a clearer determination of whether SUA is causally associated with specific neonatal outcomes. Finally, we would like to highlight the scarcity of studies comparable to ours. Therefore, our findings have the potential to contribute additional insights and improve understanding in this area.

While our study provides valuable findings, it is important to acknowledge certain limitations. We were unable to collect data that could have been really worthwhile, such as the presence of umbilical cord anomalies (length, placental insertion…) or placental disorders (placenta previa, placenta accreta, abrutpio placentae, etc.). Another weak point of this study was the lack of the postnatal confirmation of ISUA, as this could involve the risk of misclassification due to false positives or false negatives in prenatal ultrasound diagnosis. Last but not least, it should be noted that there is a significant time gap between when the ISUA cases were collected and when the control group cases were collected. This study was designed as such because of the low prevalence of ISUA (1%) and in order to have enough cases for comparison with a control group, but we are aware that this significant time gap may introduce confounding variables due to changes in clinical practices, technological advancements and population demographics over time.

Although our sample size exceeds that of some studies reported in the literature, a larger cohort might reveal additional statistically significant differences in certain parameters. Therefore, we believe that a prospective continuation of this study, with systematic data collection for future analysis, is warranted.

## 6. Conclusions

The findings of our study indicate that the presence of an isolated single umbilical artery (ISUA) is associated with a lower birth weight; however, it does not appear to be linked to a lower gestational age or increase the risk of prematurity. Additionally, ISUA is not related to a higher incidence of genetic abnormalities, structural malformations, low Apgar scores at birth, decreased umbilical cord blood pH, neonatal intensive care unit (NICU) admission or perinatal mortality.

## Figures and Tables

**Figure 1 jcm-13-07749-f001:**
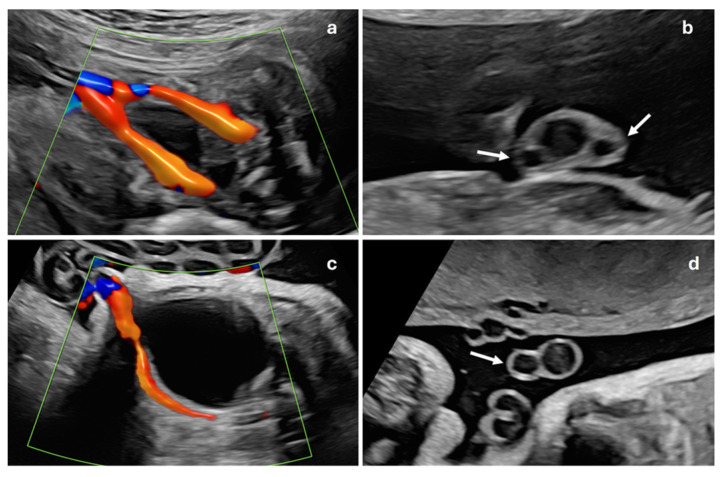
Prenatal ultrasound examination of the umbilical cord. Normal fetal umbilical cord in color Doppler mode (**a**), showing two umbilical arteries passing on either side of fetal bladder, and in grayscale transverse section (**b**), where two umbilical arteries (white arrows) and umbilical vein are visible, confirming three-vessel cord. Fetal umbilical cord with single umbilical artery in color Doppler mode (**c**), demonstrating only one umbilical artery adjacent to fetal bladder, and in grayscale transverse section (**d**), showing single artery (white arrow) and one umbilical vein, indicating two-vessel cord.

**Figure 2 jcm-13-07749-f002:**
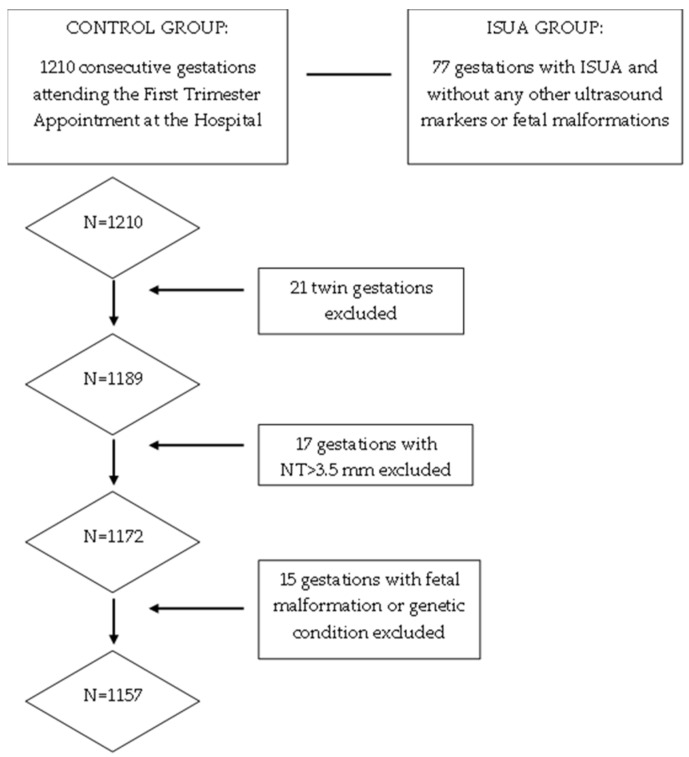
A flowchart showing the patient selection process for this study.

**Table 1 jcm-13-07749-t001:** Pregnancy characteristics and perinatal outcomes for the study and control groups.

		Three-Vessel Cord *	ISUA *	*p*
Smoking habit	No	982 (79.6)	67 (5.4)	0.57
	Yes	175 (14.2)	10 (0.8)	
Pregnancy	Spontaneous	1040 (84)	67 (5.4)	0.51
	Assisted reproductive techniques	117 (9.4)	10 (0.8)	
Labor induction	No	612 (49.6)	55 (4.4)	0.02
	Yes	545 (44.2)	22 (1.8)	
Fetal gender	Male	601 (48.7)	36 (2.9)	0.47
	Female	556 (45.1)	41 (3.3)	
APGAR 1st minute	Severe/Moderate ventilatory depression (0–6 points)	88 (7.2)	8 (0.6)	0.81
	Satisfactory status (7–10 points)	1079 (86.7)	69 (5.5)	
APGAR 5th minute	Severe/Moderate ventilatory depression (0–6 points)	78 (6.4)	8 (0.6)	0.43
	Satisfactory status (7–10 points)	1079 (87.4)	69 (5.6)	
Hospital admission	No	1128 (91.4)	77 (6.2)	0.160
	Yes	29 (2.4)	0 (0)	
Newborn death	No	1151 (93.3)	77 (6.2)	0.503
	Yes	6 (0.5)	0 (0)	

* Data presented as N and (%).

**Table 2 jcm-13-07749-t002:** Association of ISUA with gestational age at birth, birth weight and immediate postnatal pH.

	Three-Vessel Cord		ISUA		*p*
	Mean	SD	Mean	SD	
Gestational age (weeks)	39	1.8	38	4.4	0.04
Newborn weight (g)	3183.7	478.9	3013.2	507.3	0.045
pH	7.3	0.1	7.25	0.1	0.413
Maternal BMI	25.1	5.7	27	6	0.77
Maternal age	32.5	5.8	33	5.9	0.99

**Table 3 jcm-13-07749-t003:** Multivariate logistic regression analysis of factors associated with presence of ISUA.

	OR	CI (95%)	*p*
Gestational age (weeks)	1.21	0.99–1.63	0.05
Newborn weight (g)	0.99	0.98–0.99	0.05
pH	2.92	0.014–42.6	0.85
APGAR 1st minute	3.53	0.32–47.5	0.30
APGAR 5th minute	0.06	0.008–1.73	0.12

## Data Availability

The data presented in this study are available on request from the corresponding author due to the data protection of the patients involved.

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
