# Peer review of "Fetal Isolated Single Umbilical Artery (ISUA) and Its Role as a Marker of Adverse Perinatal Outcomes"

_jcm, 2024, doi:10.3390/jcm13247749_

Round 1
Reviewer 1 Report
Comments and Suggestions for Authors
Line 16- growth restriction, not retardation is the commonly accepted term for FGR.
Line 20- Research does not "check" this needs re-wording as does the next sentence which also doesn't make grammatical sense.
Line 27- Apgar is a score, not a test.
Line 31- Conclusion needs a stronger closing sentence. It currently ends with a summary of results with no actual conclusion.
Line 42- again, IUGR is incorrect. FGR.
Table 2 - reports p values as a group but these outcomes are not dichotomous, so why only 1 p value for a list of conditions?
Line 145- What is AUU?
Line 169- remove this personal comment. Interesting is up to the reader.
Comments on the Quality of English Language
Terms such as IUGR are no longer used, we use FGR. Retardation is not an acceptable term. Other sentences do not make grammatical sense.
Author Response
First of all we would like to thank reviewer number 1 for his/her kind comments and suggestions to improve our paper.
Below we will be replying to his/her suggestions one by one, as requested.
Line 16- growth restriction, not retardation is the commonly accepted term for FGR.
We would like to thank the reviewer for pointing out this mistake, which has now been corrected. The term “Intrauterine Growth Retardation” has been changed and replaced by “Fetal Growth Restriction”.
Line 20- Research does not "check" this needs re-wording as does the next sentence which also doesn't make grammatical sense.
This sentence has been changed into “This study aims to evaluate whether the presence of isolated SUA (ISUA) is related to adverse perinatal outcomes”.
Line 27- Apgar is a score, not a test.
We definitely thank the reviewer for pointing out that mistake, that has been corrected and changed into “Apgar score” in all the places where it was misspelled.
Line 31- Conclusion needs a stronger closing sentence. It currently ends with a summary of results with no actual conclusion.
We have changed the conclusion, making making the sentences clearer and more precise.
Line 42- again, IUGR is incorrect. FGR.
This item has been already answered.
Table 2 - reports p values as a group but these outcomes are not dichotomous, so why only 1 p value for a list of conditions?
Initially, we decided to divide certain variables into three groups to facilitate comparisons and identify potential differences among them. However, the preliminary analysis did not reveal any significant differences, rendering ad hoc comparisons unfeasible. Consequently, we opted to present the significance values obtained from the initial analysis. However, after reading the reviewer's advice, we recognized that this approach might lead to misinterpretation. Therefore, we have decided to reclassify the variable into only two categories and report the significance level obtained in the initial analysis, which again was not statistically significant.
Line 145- What is AUU?
Again, we thank the reviewer for pointing out that typo. AUU is the Spanish word for ISUA, and there was a mistake in the translation that has now been corrected.
Line 169- remove this personal comment. Interesting is up to the reader.
As the reviewer kindly suggested, the sentence “Finally, we would like to point out that there are few studies similar to ours, which makes our study innovative and interesting” has been deleted and changed into “Finally, we would like to highlight the scarcity of studies comparable to ours. Therefore, our findings have the potential to contribute additional insights and improve understanding in this area.”.
Reviewer 2 Report
Comments and Suggestions for Authors
The authors present a retrospective observational study aiming to assess whether the presence of isolated single umbilical artery (ISUA) is associated with adverse perinatal outcomes. The study compares 77 cases of ISUA with 1,189 normal (it is not clear if aneuploidies and congenital abnormalities have been included) pregnancies. The findings suggest that ISUA is associated with lower gestational age and birth weight but not with increased rates of prematurity, genetic disorders, fetal malformations, adverse Apgar scores, lower pH values at birth, or increased neonatal intensive care unit (NICU) admissions.
Major Concerns:
Methodology and Study Design:
- Lack of Inclusion and Exclusion Criteria: The manuscript does not specify the inclusion and exclusion criteria used to select the study population. It is essential to detail these criteria to understand how participants were selected and to assess potential selection bias. Additionally, a specific definition for ISUA and a description of the control group are needed. The only clear information from the manuscript is that they are singleton pregnancies.
- No Flowchart Provided: There is no flowchart outlining the selection process with reasons for exclusion.
- Unclear Control Group Composition: It is unclear which population the control group was drawn from. In Table 2, the control group is referred to as "no ISUA." Does the control group include non-isolated SUA cases? Does it include aneuploidies and congenital abnormalities? If so, this should be corrected, as ISUA should be compared with normal singleton pregnancies without aneuploidies or congenital abnormalities.
- Details on Ultrasound Examinations: The study lacks information on when, how many and from who (specialist?) ultrasound scans were performed, the gestational ages at which ISUA was diagnosed, and whether standardized protocols were followed. Additionally, it is unclear if the ISUA diagnosis was confirmed postnatally through examination of the umbilical cord and placenta.
- Confirmation of ISUA Diagnosis: Without confirmation of ISUA after birth, there is a risk of misclassification due to false positives or false negatives in prenatal ultrasound diagnosis. Details on postnatal verification are crucial for the validity of the study findings. Otherwise, this limitation should be mentioned among the weaknesses of the study.
- Temporal Discrepancy in Data Collection: The control group data were collected from January to August 2013, whereas the ISUA cases were collected from 2019 to 2023. This significant time gap may introduce confounding variables due to changes in clinical practices, technological advancements, and population demographics over time. This should be mentioned as a weakness.
Statistical Analysis:
- Absence of Multivariate Analysis: The study reports associations based on univariate analyses without adjusting for potential confounding factors such as maternal age, body mass index (BMI), smoking status, or assisted reproductive technologies. A multivariate analysis is preferred, if possible, to determine whether ISUA independently predicts adverse outcomes.
- Sample Size Justification: There is no mention of a power calculation or justification for the sample size. Given the relatively small number of ISUA cases (77), the study may be underpowered to detect significant differences in some outcomes.
Results Interpretation:
- Prematurity Rate: While the study notes a lower gestational age in ISUA cases, it concludes there is no increased prematurity rate. Clarification is needed on how prematurity was defined.
- Presentation of Population Characteristics: In Table 1, which includes the characteristics of the population, the characteristics of the study group (ISUA) and the control group (normal singleton pregnancies with no aneuploidies or congenital malformations) should be presented separately to make their differences visible.
Language and Grammar:
- English Proficiency: The manuscript contains grammatical errors and awkward phrasing that impede comprehension. For example, phrases like "the presence of ISUA has not been associated with genetic abnormalities, as we have not demonstrated a statistically significant association between the two variables" can be more succinctly expressed.
- Consistency and Clarity: Inconsistent use of terms (e.g., "intrauterine growth retardation" vs. "intrauterine growth restriction") and occasional typos detract from the manuscript's professionalism.
Minor Concerns:
Data Presentation: In Table 2, the p-values are referred to more than one variable; please explain what this means.
Conclusion:
The study addresses an important question regarding the impact of ISUA on perinatal outcomes. However, significant revisions are necessary to strengthen the methodology, clarify the analysis, and improve the overall quality of the manuscript. By addressing the concerns outlined above, the authors can enhance the credibility and contribution of their research to the field. No comments have been made regarding the introduction and discussion, as the above-stated serious matters should be corrected in the methodology and results first.
Comments on the Quality of English LanguageEnglish Proficiency: The manuscript contains grammatical errors and awkward phrasing that impede comprehension. For example, phrases like "the presence of ISUA has not been associated with genetic abnormalities, as we have not demonstrated a statistically significant association between the two variables" can be more succinctly expressed.
Consistency and Clarity: Inconsistent use of terms (e.g., "intrauterine growth retardation" vs. "intrauterine growth restriction") and occasional typos detract from the manuscript's professionalism.
Author Response
First of all, we would like to thank reviewer number 1 for his/her kind and accurate comments and suggestions which we believe have contributed to improving our article. In line with this, the manuscript has been extensively revised and virtually fully rewritten to incorporate theinsightful changes suggested by the reviewer.
Below we will be replying to his/her suggestions one by one, as requested.
Lack of Inclusion and Exclusion Criteria: The manuscript does not specify the inclusion and exclusion criteria used to select the study population. It is essential to detail these criteria to understand how participants were selected and to assess potential selection bias. Additionally, a specific definition for ISUA and a description of the control group are needed. The only clear information from the manuscript is that they are singleton pregnancies.
The information requested by the reviewer has been included. The criteria for both inclusion and exclusion in the study have been clarified and explained. Furthermore, on review, and taking into account point number 3 of his7her suggestions, cases with congenital anomalies or aneuploidies have been excluded from the control group, leaving this group only with singleton gestations that are fully normal. The definition of ISUA has also been clarified, as requested.
No Flowchart Provided: There is no flowchart outlining the selection process with reasons for exclusion.
Following the reviewer's indications, a flowchart showing the selection process has been made, which is shown as Figure 1.
Unclear Control Group Composition: It is unclear which population the control group was drawn from. In Table 2, the control group is referred to as "no ISUA." Does the control group include non-isolated SUA cases? Does it include aneuploidies and congenital abnormalities? If so, this should be corrected, as ISUA should be compared with normal singleton pregnancies without aneuploidies or congenital abnormalities.
We found this suggestion really interesting and for this reason we reconsidered the initial selection of the patients. We have modified and corrected this point, as suggested by the reviewer, so that the control group is now composed only by singleton pregnancies with normal course (i.e. without the presence of aneuploidies or congenital abnormalities). For this reason, the sample size of the control group has been modified, as well as the statistical analysis.
Details on Ultrasound Examinations: The study lacks information on when, how many and from who (specialist?) ultrasound scans were performed, the gestational ages at which ISUA was diagnosed, and whether standardized protocols were followed. Additionally, it is unclear if the ISUA diagnosis was confirmed postnatally through examination of the umbilical cord and placenta.
This point has also been clarified, following the reviewer's recommendation.
The Prenatal Diagnostic Department of our hospital is composed by five doctors (AMC, MOMR, PO, AVY, FJG) who follow specific internationally accepted and standardized protocols, the ISUOG protocols. This has now been stated in the paper and the reference of the protocol has been included in the reference list.
Unfortunately, a postnatal confirmation of the presence of a single umbilical artery in the cord was not performed, nor was the placenta examined, which would have been really useful. As the reviewer stated, this point has been identified as one of the weaknesses of the study.
Confirmation of ISUA Diagnosis: Without confirmation of ISUA after birth, there is a risk of misclassification due to false positives or false negatives in prenatal ultrasound diagnosis. Details on postnatal verification are crucial for the validity of the study findings. Otherwise, this limitation should be mentioned among the weaknesses of the study.
This question has been answered in the previous point.
Temporal Discrepancy in Data Collection: The control group data were collected from January to August 2013, whereas the ISUA cases were collected from 2019 to 2023. This significant time gap may introduce confounding variables due to changes in clinical practices, technological advancements, and population demographics over time. This should be mentioned as a weakness.
Following the reviewer’s suggestions, a paragraph addressing this point has been added to the section on the study's limitations.
Statistical Analysis:
Absence of Multivariate Analysis: The study reports associations based on univariate analyses without adjusting for potential confounding factors such as maternal age, body mass index (BMI), smoking status, or assisted reproductive technologies. A multivariate analysis is preferred, if possible, to determine whether ISUA independently predicts adverse outcomes.
We do appreciate the reviewer’s thoughtful suggestion on this matter.
Initially, we did not perform multivariate analysis, as no significant differences were identified in the initial univariate analysis for any of the variables. To address this point, statistical assistance was requested from the Department of Statistics at the Faculty of Medicine (Dr. Mercedes Sánchez Barba, now included as an author in the article). Based on her recommendation, an artificial intelligence algorithm (Random Forest method) was employed to improve the assessment given the disparity in the sizes of the two study groups (control and ISUA). The conclusions are detailed in the article: no association or modification of any of the variables included in the multivariate analysis was identified using the Random Forest method.
Sample Size Justification: There is no mention of a power calculation or justification for the sample size. Given the relatively small number of ISUA cases (77), the study may be underpowered to detect significant differences in some outcomes.
Again, thanks to the reviewer for this valuable recommendation. A paragraph explaining the sample size calculation process has been added. Sample size was calculated using the WinEpi calculator (http://www.winepi.net/f102.php) considering a prevalence of 1% for ISUA with a confidence level of 95%.
Results Interpretation:
Prematurity Rate: While the study notes a lower gestational age in ISUA cases, it concludes there is no increased prematurity rate. Clarification is needed on how prematurity was defined.
Thank you once again for this insightful suggestion. We have added the WHO definition of prematurity to this paragraph and cited the corresponding reference in the bibliography, so that the text is now as follows: “Despite this reduction in gestational age, the preterm birth rate was not higher, adhering to the WHO definition of preterm birth as delivery before 37 weeks of gestation”.
Presentation of Population Characteristics: In Table 1, which includes the characteristics of the population, the characteristics of the study group (ISUA) and the control group (normal singleton pregnancies with no aneuploidies or congenital malformations) should be presented separately to make their differences visible.
Following Reviewer’s suggestion, tables have been modified and the characteristics of the population have been presented separately, divided into the characteristics of the study group (ISUA) the control group (three vessel cord).
Language and Grammar:
English Proficiency: The manuscript contains grammatical errors and awkward phrasing that impede comprehension. For example, phrases like "the presence of ISUA has not been associated with genetic abnormalities, as we have not demonstrated a statistically significant association between the two variables" can be more succinctly expressed.
We deeply regret this inconvenience. Following the reviewer’s suggestion, A thorough revision of the manuscript's grammar and language has been undertaken to ensure clarity and improve readability. We hope this has been successfully achieved.
Consistency and Clarity: Inconsistent use of terms (e.g., "intrauterine growth retardation" vs. "intrauterine growth restriction") and occasional typos detract from the manuscript's professionalism.
We thank the reviewer again for pointing out these mistakes. These words have been corrected and properly changed.
Minor Concerns:
Data Presentation: In Table 2, the p-values are referred to more than one variable; please explain what this means.
We would like to explain this issue. Initially, we decided to divide certain variables into three groups to facilitate comparisons and identify potential differences among them. However, the preliminary analysis did not reveal any significant differences, rendering ad hoc comparisons unfeasible. Consequently, we opted to present the significance values obtained from the initial analysis. However, after reading the reviewer's advice, we recognized that this approach might lead to misinterpretation. Therefore, we have decided to reclassify the variable into only two categories and report the significance level obtained in the initial analysis, which again was not statistically significant.
Conclusion:
The study addresses an important question regarding the impact of ISUA on perinatal outcomes. However, significant revisions are necessary to strengthen the methodology, clarify the analysis, and improve the overall quality of the manuscript. By addressing the concerns outlined above, the authors can enhance the credibility and contribution of their research to the field. No comments have been made regarding the introduction and discussion, as the above-stated serious matters should be corrected in the methodology and results first.
Round 2
Reviewer 2 Report
Comments and Suggestions for Authors
The paper presents significant findings, but there are a few areas where improvements would enhance clarity and statistical rigor.
1. Table 1: Adding Percentages
Table 1 currently lacks percentages, which would be helpful for better interpretation of the data. I recommend adding percentages where applicable to make the information more accessible and understandable for the readers.
2. Table 2: Statistical Analysis for Normal Variables
In Table 2, all of the variables are normally distributed. Therefore, it would be more appropriate to report the mean and standard deviation (SD) rather than the median. This adjustment would provide a clearer representation of the data’s central tendency and variability.
3. Random Forest and Multivariable Analysis
The random forest model used in the paper does not appear to be the correct method for multivariable analysis in this context. A multiple linear or logistic regression (depending on the type of variable under investigation) would be more appropriate to examine the relationships between the variables and the outcome. I recommend replacing the random forest model with the appropriate regression analysis.
Author Response
The paper presents significant findings, but there are a few areas where improvements would enhance clarity and statistical rigor.
- Table 1: Adding Percentages
Table 1 currently lacks percentages, which would be helpful for better interpretation of the data. I recommend adding percentages where applicable to make the information more accessible and understandable for the readers.
We thank the reviewer for this comment, which undoubtedly enhances the interpretation of the data. Following his/her suggestion, percentages have been added to Table 1.
- Table 2: Statistical Analysis for Normal Variables
In Table 2, all of the variables are normally distributed. Therefore, it would be more appropriate to report the mean and standard deviation (SD) rather than the median. This adjustment would provide a clearer representation of the data’s central tendency and variability.
We apologize for this typographical error: "media" means "mean" in Spanish, and its appearance in the table was a typo. It has now been corrected in table 2.
- Random Forest and Multivariable Analysis
The random forest model used in the paper does not appear to be the correct method for multivariable analysis in this context. A multiple linear or logistic regression (depending on the type of variable under investigation) would be more appropriate to examine the relationships between the variables and the outcome. I recommend replacing the random forest model with the appropriate regression analysis.
We would like to say that the Random Forest model was suggested by the statistics department of the faculty as an AI-assisted model to improve statistical power, considering the highly asymmetric size of the study groups. This was the reason for its use. However, following the reviewer’s suggestion, it has been replaced with a multivariate logistic regression analysis in which confidence intervals are presented. In this analysis, although statistical significance is achieved for gestational age at birth in both univariate and multivariate analysis, the confidence interval includes 1, so the results and conclusions have been modified, considering the association between ISUA and lower gestational age at birth as non-significant.